# Why Diffusion Models Are Stable and How to Make Them Faster: An Empirical Investigation and Optimization

## Abstract

Diffusion models, a potent generative framework, have garnered considerable attention in recent years. While many posit that the superiority of diffusion models stems from their stable training process compared to Generative Adversarial Networks (GANs), these assertions often rest on intuition and lack empirical substantiation. In this paper, we aim to provide direct evidence to explain why diffusion models exhibit remarkable stability during training. We start by comparing the generation results of models with different hyper-parameters, such as initialization and model structure, under the same sampling conditions. Our results show that diffusion models produce consistent generation results across different hyper-parameters, indicating that they are stable in learning the mapping between noise and data. We then compare the loss landscapes of diffusion models and GANs, and find that diffusion models have much smoother loss landscapes, implying better convergence stability. Based on these analyses, we propose two optimization methods for diffusion models, namely the curriculum learning based timestep schedule (CLTS) and the momentum decay with learning rate compensation (MDLRC), which optimize the sampling probability of timesteps and the momentum selection, respectively, to accelerate convergence. For example, on ImageNet128, our methods achieve a 2.6x speedup in training, demonstrating the effectiveness of our methods.

## 1 Introduction

Diffusion Models (DMs) Sohl-Dickstein et al. (2015); Ho et al. (2020); Song et al. (2020b;a), a prominent class of generative models, have garnered considerable attention in recent years. Owing to their exceptional capability to model intricate data distributions, DMs have catalyzed significant advancements in various domains. These include image generation Nichol & Dhariwal (2021); Dhariwal & Nichol (2021); Rombach et al. (2021), image manipulation Zhang et al. (2023b); Lugmayr et al. (2022); Kawar et al. (2023), video generation Ho et al. (2022); Blattmann et al. (2023); Wang et al. (2023), and speech synthesis Jeong et al. (2021); Zhang et al. (2023a). While the superiority of diffusion models is often attributed to their stable training process Ho et al. (2020), these claims are frequently based on intuition and lack empirical evidence.

In this study, we endeavor to provide empirical evidence substantiating the stability of the training process in DMs. Based on our findings, changing the model structure and initialization would not significantly influence the result as long as training with constant noise, *i.e.*, the same initial noise and noise per round. We try to reveal the veil of the stability of DMs from the perspective of the training landscape. Our results reveal a notable consistency in the generative outcomes, as depicted in Figure 1. It is important to highlight that such a consistency phenomenon is not typically observed in generative models. These models generally bootstrap samples that adhere to a certain probability distribution, *i.e.,* a noise, onto the desired data distribution in an ultra-high dimensional space Song & Ermon (2019). This process is inherently laden with a considerable degree of randomness. Consequently, this experiment demonstrates the stability of DMs in learning noise-data mapping relationships and the robustness of hyper-parameters.

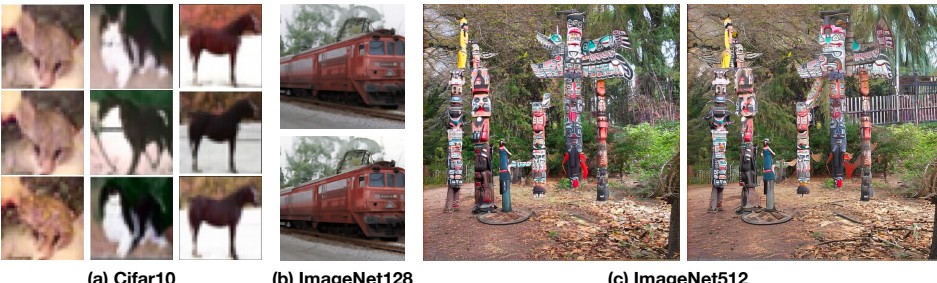

**(a) Cifar10**  **(b) ImageNet128**  **(c) ImageNet512**

Figure 1: Illustration of the consistency phenomenon in diffusion models (DMs). Despite different initializations or structural variations, DMs trained on the same dataset produce remarkably consistent results when exposed to identical noise during sampling. (a) presents three models Nichol & Dhariwal (2021) trained on CIFAR10 with different initializations. (b) depicts two models Dhariwal & Nichol (2021) trained on ImageNet128 with different structures. (c) showcases the large and huge models of UViT Bao et al. (2023) trained on ImageNet512.

From the observation of DMs mentioned above, we can reasonably speculate that *The landscape of DMs resembles a bowl*, which implies that the model from different initialization and structure converges to a similar minimum. To further investigate the loss landscape associated with DMs, we employ various techniques such as loss landscape visualization Li et al. (2018), 1D interpolation Li et al. (2018), and Hessian spectral decomposition Yao et al. (2020). Loss landscape is the high-dimensional space formed by the partial derivatives of the loss function with respect to the model parameters Li et al. (2018). The smoothness of this space affects the convergence rate of the model Garrigos & Gower (2023), *i.e.,* the smoother the loss landscape, the easier the optimization. We compare the results of these techniques on DMs and a conventional generative model, *e.g.,* GANs, in Fig. 2, Fig. 3 and Fig. 4, respectively. Our findings reveal that the loss landscape of the diffusion model exhibits significantly higher smoothness compared to that of GANs, implying that DMs are easier to optimize than GANs.

Motivated by these analyses, we propose two optimization approaches for DMs. First, we further examine the consistency phenomenon of DMs and discover that different timesteps have different convergence difficulties and their contribution to the final generation quality varies Choi et al. (2022). Therefore, we propose the curriculum learning based timestep schedule Bengio et al. (2009) (CLTS), which aims to gradually decrease the sampling probability of easy-to-converge timesteps and thus improve training efficiency. Second, we propose momentum decay with learning rate compensation (MDLRC), which exploits the high smoothness of the loss landscape of the diffusion model. Unlike GANs, which require a large momentum to ensure gradient stability, the diffusion model can benefit from a smaller momentum and a higher learning rate, which can further improve the convergence efficiency.

We evaluate our optimization methods on various DMs, such as Nichol & Dhariwal (2021) and Dhariwal & Nichol (2021), and demonstrate that they can enhance the convergence speed of DMs. For instance, on ImageNet128 Deng et al. (2009), our methods achieve a 2.6× speedup in training with standard Guided Diffusion Dhariwal & Nichol (2021) and a 2× speedup on CIFAR10 Krizhevsky et al. (2009) compared with standard Improved Diffusion Nichol & Dhariwal (2021), showing the effectiveness of our methods. In summary, our main contributions are three-fold:

- We provide empirical evidence to explain why diffusion models exhibit remarkable stability during training, by conducting a consistency experiment and comparing the loss landscapes of diffusion models and GANs.

- We propose two optimization methods for diffusion models, namely the curriculum learning based timestep schedule (CLTS) and a momentum decay with learning rate compensation (MDLRC), which optimize the sampling probability of timesteps and the momentum selection, respectively, to accelerate model convergence.

- We evaluate our optimization methods on various diffusion models and datasets, and demonstrate their effectiveness in enhancing the convergence speed of diffusion models.

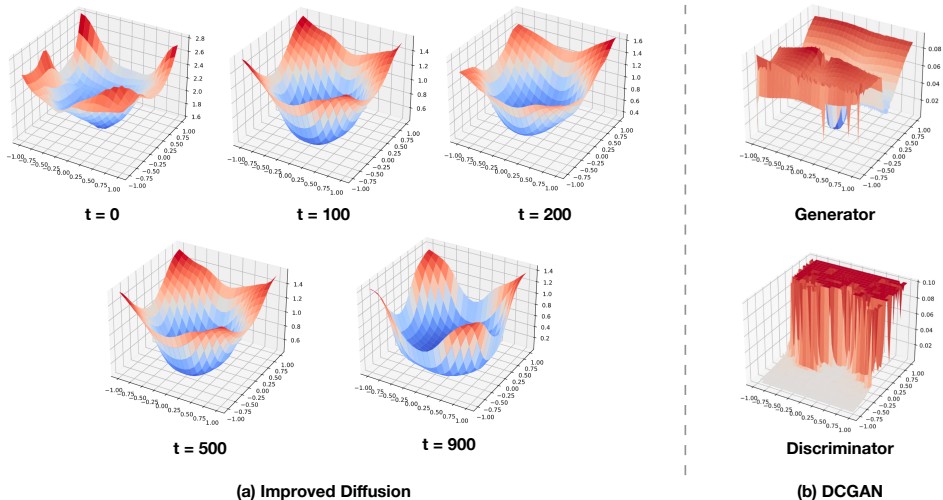

Figure 2: Visualization of the loss landscapes of Improved Diffusion Nichol & Dhariwal (2021) and DCGAN Radford et al. (2015), where **t** is the timestep of DMs. Both models were trained on the CIFAR10 dataset Krizhevsky et al. (2009). Obviously, the loss landscape of DMs is smoother compared to GANs. More landscapes of DM and GAN can be viewed at Appendix D

## 2 RELATED WORK

### 2.1 DIFFUSION MODELS

Diffusion Models (DMs) are a class of generative models that use techniques from non-equilibrium thermodynamics to learn the latent structure of complex data distributions. They were first introduced by Sohl-Dickstein et al. (2015), who applied their method to image and text generation. Later, Ho et al. (2020) proposed Denoising Diffusion Probabilistic Models (DDPM), which improved the sampling efficiency and quality of diffusion models by using a denoising score matching objective and a learned diffusion process. Rombach et al. (2021) developed Latent Diffusion Models (LDMs), which compressed high-resolution images into lower-dimensional representations using pretrained autoencoders. They also introduced cross-attention layers into the model architecture, which enabled LDMs to handle various conditioning inputs, such as text or bounding boxes, and generate high-resolution images in a convolutional manner. Despite the success of DMs using UNet Ho et al. (2020); Nichol & Dhariwal (2021); Rombach et al. (2021), a convolutional neural network, Bao et al. (2023) and Peebles & Xie (2022) discovered the feasibility of using Vision Transformer Dosovitskiy et al. (2020) in DMs, achieving state-of-the-art generation results.

Several studies have focused on improving the DMs from various aspects Karras et al. (2022); Chen (2023). Nichol & Dhariwal (2021) proposed several techniques to enhance the performance and efficiency of DMs, such as employing a learned variance schedule, adopting a cosine timestep schedule for low-resolution data, and developing a multi-scale architecture. Dhariwal & Nichol (2021) further improved the performance and fidelity of DMs, by incorporating advanced design concepts of BigGAN Brock et al. (2018). Although integrating the sophisticated model structure of GAN can benefit the performance of DMs, they also adopt the same large momentum setting, which is suboptimal, because the loss landscape of DMs is highly smoothed. A large momentum not only affects convergence efficiency but also causes oscillations. We discussed this in detail in Section 3.2.

### 2.2 TRAINING ACCELERATION

Learning from easy to hard is the core concept of curriculum learning Bengio et al. (2009), *e.g.*, Platanios et al. (2019) Curriculum Learning improves language translation model the performance by up to 2.2 and reduces training time by up to 70%. Soviany et al. (2020) trains a support vector regression model to assess the difficulty of images, and subsequently, training with the selected images

results in improved stability and superior performance. Ghasedi et al. (2019) proposes a heuristic strategy to assign weights to individual images during the training process.

Additionally, sparse training is an effective technique for enhancing network performance and accelerating training speed Mi et al. (2023). Mi et al. (2023) applies a series of hardware-friendly sparse masks to the variables during the forward and reverse processes to achieve acceleration. Another option is to consider acceleration from the perspective of an optimizer Gupta et al. (2018); Martens & Grosse (2015); Frantar et al. (2021); Liu et al. (2023). Frantar *et al.* Frantar et al. (2021) presents a highly efficient algorithm for approximating matrices, specifically designed for utilization in second-order optimization.

Momentum decay is another promising technique for accelerating the training process. Chen & Kyrillidis (2019) introduced Decaying Momentum (Demon), a method that reduces the cumulative effect of a gradient on all future updates. Demon outperformed other methods in 28 different settings, involving various models, epochs, datasets, and optimizers. This work inspired our proposed optimization method.

Recently, motivated by the large computation cost in training DMs, several approaches focus on training acceleration of DMs Hang et al. (2023); Wu et al. (2023); Choi et al. (2022). Hang et al. (2023) treat the diffusion training as a multi-task learning problem, and introduce a simple yet effective approach called Min-SNR, which adapts the loss weights of timesteps based on clamped signal-to-noise ratios. Wu et al. (2023) proposed the momentum-based diffusion process, which can be modeled as a damped oscillation system, whose critically damped state has the optimal noise perturbation kernel that avoids oscillation and accelerates the convergence speed.

## 3 UNVEILING THE STABILITY OF DIFFUSION MODELS

In this section, we present empirical evidence to substantiate the stability of DMs in learning noise-to-data mapping and convergence, thereby underscoring their superiority over GANs. We delve into an analysis of the stability of DMs in the context of learning noise-to-data mapping in Section 3.1. Finally, we draw a comparison between the smoothness of the loss landscape of DMs and GANs in Section 3.2. We revisited the formulation of DMs in Appendix A.

### 3.1 ANALYZE THE STABILITY OF NOISE-TO-DATA MAPPING

The stability of the generative model learning the noise-to-data mapping is an important aspect of generative modeling, as it reflects how well the model can cope with different noise and different choices of hyper-parameters. However, this stability is often overlooked or not explicitly evaluated.

In this section, we evaluate the stability of DMs in learning the noise-to-data mapping through a consistency experiment. We select three diffusion frameworks as representative DMs: Improved Diffusion Nichol & Dhariwal (2021), Guided Diffusion Dhariwal & Nichol (2021), and U-ViT Bao et al. (2023), and two GAN frameworks: DCGAN Radford et al. (2015) and BigGAN Brock et al. (2018). We train DMs and GANs with different hyper-parameters on three benchmarks: Cifar10 Krizhevsky et al. (2009), ImageNet128 Deng et al. (2009) and ImageNet512 Deng et al. (2009). The results of the consistency experiment and the detailed settings are presented in Table 1. The generated images are shown in Appendix E.

In the consistency experiment, we use the peak signal-to-noise ratio (PSNR) to measure the consistency of each model. Specifically, for a group of models, *e.g.,* different initializations of Improved Diffusion or large and huge models of U-ViT, we sample 32 images with the same sampling seed, which means the initial noise and the noise per round are the same. Suppose we have $N$ models of a group, each model generate $M$ images, we then measure the consistency $C(\cdot)$,

$$C(q) = \frac{1}{M} \sum_{i=1}^{M} \frac{1}{N-1} \sum_{j=2}^{N} \text{PSNR}(q_{i,1}, \ q_{i,j}), \tag{1}$$

where $q \in \mathbb{R}^{N \times M}$ is the matrix of images.

Table 1: Comparing consistencies of DMs and GANs in learning noise-to-data mapping.

| | Datasets | Different Initializations | Different Structures | Consistency (PSNR) |
|---|---|---|---|---|
| Improved Diffusion | Cifar10 | ✓ | | **20.14** |
| DCGAN | Cifar10 | ✓ | | 10.48 |
| Guided Diffusion | ImageNet128 | ✓ | ✓ | **17.23** |
| BigGAN | ImageNet128 | ✓ | ✓ | 8.58 |
| U–ViT | ImageNet512 | ✓ | ✓ | **14.37** |
| BigGAN | ImageNet512 | ✓ | ✓ | 6.40 |

The result of the consistency experiment reveals that all DMs have much higher consistency than GANs, regardless of the dataset, initialization, or model structure, indicating that DMs are more robust and stable in learning noise-to-data mapping.

## 3.2 ANALYZE THE SMOOTHNESS OF LANDSCAPE

The smoothness of the loss landscape is strongly correlated with the convergence difficulty Li et al. (2018). In this section, we conduct a thorough investigation of the loss landscape of DMs and GANs during the training. However, due to the high dimensionality of the models' parameters, it is infeasible to access the full information of the loss landscape. Therefore, we resort to a partial analysis based on 1D interpolation of models and hessian spectra, following the method proposed by Li et al. (2018).

**1D interpolation** is a technique that generates new data points by leveraging existing data. In our research, we employed 1D linear interpolation to estimate the position $\boldsymbol{\theta}$ within the landscape using the provided models at different stages of training, namely $\boldsymbol{\theta}_a$ and $\boldsymbol{\theta}_b$. This involved calculating the weighted sum of these two models.

$$\boldsymbol{\theta} = \alpha\boldsymbol{\theta}_a + (1-\alpha)\boldsymbol{\theta}_b. \quad (0 \leq \alpha \leq 1) \tag{2}$$

We use interpolation to analyze the relationship between different training stages and gather valuable information. Our approach involves training a Diffusion model and a GAN model, followed by selecting models from various training steps as anchor points. Specifically, we select the models trained 10 and 100 epochs for both DM and GAN. These selections, shown in Figure 3, represent models from both early and late convergence stages.

As shown in Figure 3, the GAN model exhibits more erratic changes in loss, indicating that changes in GAN parameters lead to relatively larger changes.

**Hessian spectra** refers to the distribution of eigenvalues in the Hessian matrix. Inspired by the connection between the geometry of the loss landscape and the eigenvalue, we approximate the Hessian spectrum by the Lanczos algorithm and the results of Diffusion and GAN are shown in Figure 4. From the figure, it can be seen that the dominant eigenvalue of GAN's is larger, *i.e.*, $\lambda_1 = 13.3$(DMs) *v.s.* $\lambda_1 = 40.9$(GANs), and dispersion, *i.e.*, $\sigma^2 = 37.9$(DMs) *v.s.* $\sigma^2 = 93.9$(GANs), which implies that the landscape of GAN is steeper and more rugged, which also means that the GAN is more difficult to optimize.

## 4 OPTIMIZATION

In this section, we first uncover the cause of the unique consistency phenomenon (in Section 4.1) that we have observed only in DMs. We show that the $\epsilon$-predicted DMs tend to be trivial when timesteps $t \rightarrow T$, which leads to high similarity in structure but low diversity in detail among the generated images. We then discuss how we can exploit this property to optimize the DMs, inspired by curriculum learning Bengio et al. (2009). We propose a timestep schedule that gradually decreases the sampling probabilities of timesteps $t \rightarrow T$ as the training progresses (in Section 4.2). Additionally, in Section 4.3, we provide details of our optimal momentum schedule based on the highly smoothed loss landscape of DMs.

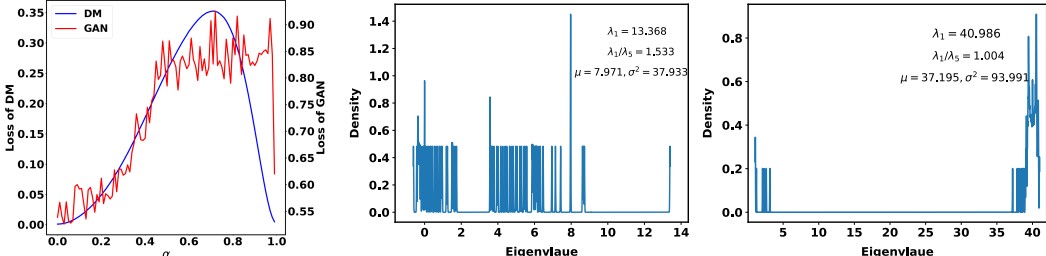

Figure 3: Illustration of the 1D-interpolation results of DMs and GANs. The jitter red line indicates the geometry of GAN's landscape is rougher.

Figure 4: Illustration of the Hessian spectrum of DMs(left) and GANs(right). $\lambda_i$ is the $i$-th largest eigenvalue and $\mu$ and $\sigma$ is the mean and variance of eigenvalue respectively. Larger dominant eigenvalue, sharper the landscape, and the greater the differences among eigenvalues, the more difficult the model is to optimize.

## 4.1 UNCOVERING THE CONSISTENCY PHENOMENON

We start with the formulation of the forward diffusion process, derived from Eq. 11,

$$x_t = \sqrt{\bar{\alpha}_t} \cdot x_0 + \sqrt{1 - \bar{\alpha}_t} \cdot \epsilon, \ \epsilon \in \mathcal{N}(0, \mathbf{I}), \tag{3}$$

where $\alpha_t = 1 - \beta_t$ and $\bar{\alpha}_t = \Pi_{s=0}^{t} \alpha_s$. Note that, $\bar{\alpha}_t$ is a factor, ranging from 0 to 1, when $t \to T$, $\bar{\alpha}_t \to 0$ then $x_t \to \epsilon$. According to Eq. 14,

$$L_{\text{simple}} = E_{x_0 \sim q(x_0), \epsilon \sim \mathcal{N}(0, \mathbf{I})} \left[ ||\epsilon - \epsilon_\theta(x_t, t)||^2 \right], \tag{4}$$

when $x_t \to \epsilon$,

$$\epsilon_\theta \to \mathbf{I}, \tag{5}$$

which means the $\epsilon$-predicted DMs tend to be trivial when $t \to T$.

The observation of the results of the consistency experiments supports the above derivation. We find that the images (Fig. 1) with consistency phenomenon are highly similar in structure but different in detail, while the timesteps when the diffusion model generates structural information are $t \to T$ (Fig. 8).

To further validate the cause of the consistency phenomenon is the $\epsilon$-predicted mechanism leading to model triviality when $t \to T$. We train an $x_0$-predicted DM as a counterexample, we change the loss function as

$$L_{x_0} = E_{x_0 \sim q(x_0)} \left[ ||x_0 - \mu_\theta(x_t, t)||^2 \right]. \tag{6}$$

As shown in Fig. 9, the consistency phenomenon disappears. Thus, we conclude that the cause of the consistency phenomenon is the $\epsilon$-predicted mechanism leading to model triviality when $t \to T$.

## 4.2 OPTIMIZING SAMPLING PROBABILITIES OF TIMESTEPS IN TRAINING

The consistency phenomenon indicates that the timesteps $t \to T$ of DMs are easy to converge. Therefore, we propose a novel approach to improve the training efficiency of DMs. We observe that the mainstream diffusion frameworks treat every timestep equally and use uniform probability $U(t) = 1/T$ to sample timesteps during the training, which leads to redundant training for $t \to T$.

To address this issue, we adopt curriculum learning Bengio et al. (2009), a training acceleration technique that is based on the principle of learning from easy to hard. Coincidentally, DMs have the natural ability to produce data with varying levels of difficulty. Specifically, the difficulty of $\epsilon$-predicted DMs increases as the timestep decreases.

We propose the curriculum learning based timestep schedule (CLTS), which aims to gradually decrease the sampling probabilities of timesteps $t \to T$ as the training progresses and increase the probabilities of others, *i.e.,* find an optimal timestep distribution. For simplicity, we assume that the optimal timestep distribution follows a Gaussian distribution $N(\cdot)$, where the mean $\mu$ indicates the most important timesteps and others are less important,

$$N(t) = \frac{1}{\sigma\sqrt{2\pi}} \exp\left(-\frac{(t - \mu)^2}{2\sigma^2}\right). \tag{7}$$

To further reduce the number of hyper-parameters in CLTS, we set the variance $\sigma = T$, which means the distribution has a standard variance across timesteps.

Our initial idea was to shift the Gaussian distribution as the mean moves from $T$ to $0$, but this achieved little improvement. We conjecture that this is because the generation of the DMs requires the involvement of all the timesteps. The strategy of shifting the distribution leads to the sampling probabilities of all timesteps except for $t \to T$ to be too small at the initial stage. Thus, we propose a mixed distribution, introducing a factor $\gamma$ to transfer the distribution from uniform to Gaussian,

$$P(t) = (1 - \gamma)U(t) + \gamma N(t), \; \gamma = \frac{\text{current iteration}}{\text{target iteration}}, \quad (8)$$

where the target iteration is a hyper-parameter that adjusts the speed of the Gaussian distribution emerging.

It is worth noting that our proposed CLTS has a similar implementation to Hang et al. (2023) and Choi et al. (2022). However, there are significant differences in our underlying philosophies. Inspired by curriculum learning, our method is based on the principle of learning from easy to hard, while Hang et al. (2023) and Choi et al. (2022) are focused on finding an optimal distribution. Our method is much more robust and efficient in extensive experiments (we show in the next section).

### 4.3 OPTIMIZATION OF MOMENTUM SCHEDULE

Dhariwal & Nichol (2021) was a pioneering work on DMs, incorporating advanced design concepts of BigGAN Brock et al. (2018) to improve the performance and fidelity of DMs. However, they also adopted the large momentum setting, *e.g.,* $\beta_1 = 0.9$, which is sub-optimal for DMs, whose loss landscape is highly smoothed. Applying a large momentum value in DMs would not only affect convergence efficiency but also cause oscillations, *e.g.,* in Fig. 5, the baseline model suffers oscillation in steps from 2.5M to 4M.

To address this issue, we propose the momentum decay with learning rate compensation (MDLRC). Following Chen & Kyrillidis (2019), the formulation of momentum decay is

$$\beta_t = \beta_0 \cdot \frac{1 - \tau}{(1 - \beta_0) + \beta_0(1 - \tau)}, \; \tau = \frac{\text{current iteration}}{\text{total iteration}}, \quad (9)$$

where $\beta_0$ is the initial momentum factor. Note that, because the total iteration is fixed, $\tau$ is not a hyper-parameter. However, since DMs usually use Exponential Moving Average (EMA) van den Oord et al. (2017) to achieve better performance, simply applying momentum decay Chen & Kyrillidis (2019) in DMs would over-amplification the weight of the current model in EMA, thus affecting the stability of the EMA model (we prove this in Appendix C). Therefore, we compensate for the learning rate $l$:

$$l_t = l_0 \cdot \frac{1 - \beta_0}{1 - \beta_t}, \quad (10)$$

where $l_0$ is the initiate learning rate, and $(1-\beta_0)$ and $(1-\beta_t)$ denote the factor of the initiate gradient and the factor of the current gradient, respectively. This compensation can ensure the weight of the current model in EMA remains consistent.

## 5 EXPERIMENTS

In this section, we integrate our optimization methods into various DMs, including Improved Diffusion Nichol & Dhariwal (2021) and Guided Diffusion Dhariwal & Nichol (2021), trained on Cifar10 Krizhevsky et al. (2009) and ImageNet128 Deng et al. (2009), respectively. Detailed settings are shown in Appendix B.

### 5.1 ABLATIONS

To evaluate the effectiveness of our proposed methods, we performed an ablation study. Fig. 6 shows the results of the ablation study. Fig. 6 (a) compares the performance of our proposed methods with and without each module, namely momentum decay (MD), learning rate compensation (LRC), and

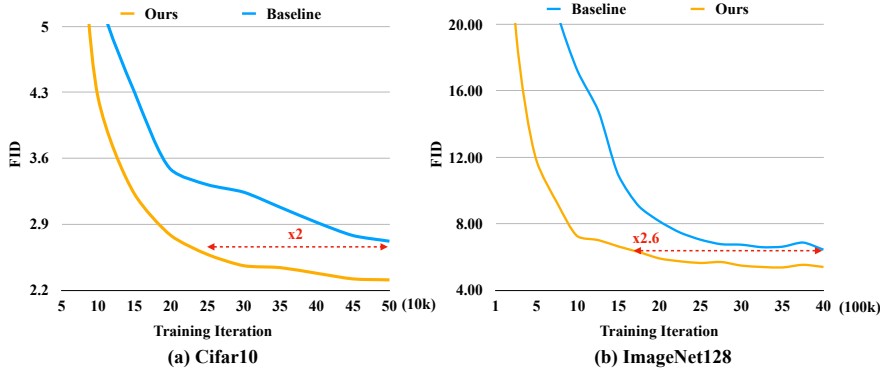

Figure 5: Illustration of the application of our optimization approach on different DMs. (a) Improved Diffusion Nichol & Dhariwal (2021) trained on Cifar10 Krizhevsky et al. (2009), (b) Guided Diffusion Dhariwal & Nichol (2021) trained on ImageNet128 Deng et al. (2009). With our methods, these DMs achieve 2× and 2.6× speedup in training, respectively.

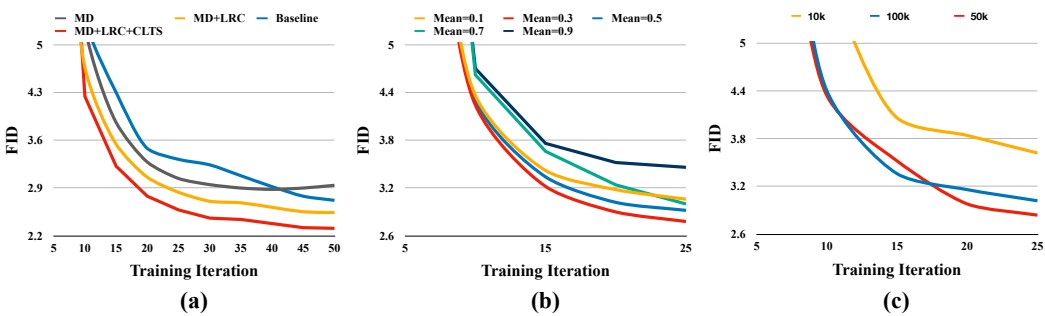

Figure 6: Ablation study, every model is trained on Cifar10. (a) shows the contributions of each module in our proposed methods. We compare the baseline model with momentum decay (MD), MD with learning rate compensation (LRC), and MDLRC with curriculum learning based timestep schedule (CLTS). (b) illustrates the influence of different mean $\mu$ in our proposed CLTS (Eq. 7). (c) reflects the influence of values of different target iterations that we used in CLTS (Eq. 8).

curriculum learning based timestep schedule (CLTS). The performance is measured by the FID score Heusel et al. (2017), which is a widely used metric for assessing the quality and diversity of generated images. The results indicate that each module enhances the performance of the model, and the combination of all modules achieves the best FID score. Fig. 6 (b) examines the effect of different mean values $\mu$ in our proposed CLTS (Eq. 7). The mean value $\mu$ controls the most important timesteps in the Gaussian distribution. The results suggest that the optimal value of $\mu$ is around 0.3, which implies that the timesteps with the highest contribution to generation are not necessarily the most difficult ones to learn. Fig. 6 (c) investigates the effect of different target iterations in our proposed CLTS (Eq. 8). The target iteration is a hyper-parameter that adjusts the speed of the Gaussian distribution emerging. The results demonstrate that the optimal value of the target iteration is around 100k, which means that the model needs about 100k iterations to fully adapt to the Gaussian distribution.

Based on the optimal settings, we trained our optimized models on Cifar10 Krizhevsky et al. (2009) and ImageNet 128 Deng et al. (2009), and compared them with the baseline models. Fig. 5 illustrates the results. The results reveal a significant acceleration of our optimized models, *e.g.,* on Cifar10, our model achieves a 2× speedup compared with the baseline model, and on ImageNet128, our model achieves a 2.6× acceleration. Fig. 7 shows the visualization results of our methods, with 1.6M training iteration, our model shows competitive visual quality with the baseline models, which has trained 4M iteration. These results demonstrate the effectiveness and robustness of our proposed methods.

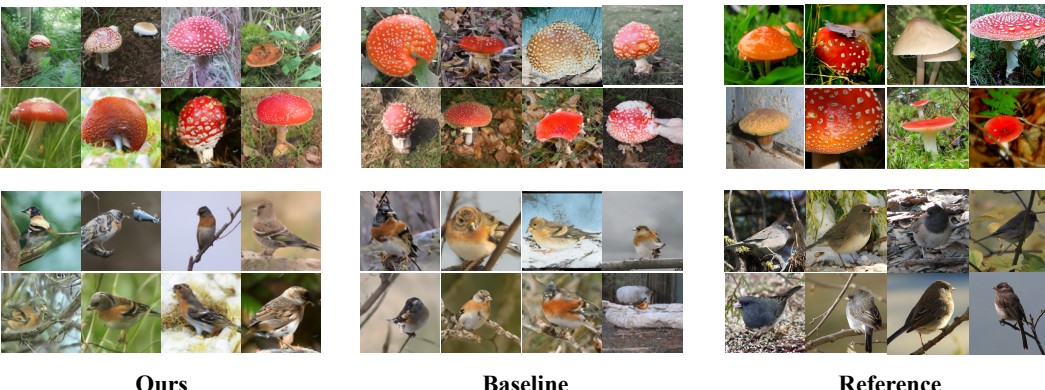

**Ours**  **Baseline**  **Reference**

Figure 7: Comparisons of generated images. Both ours and the baseline are trained on Guided Diffusion Dhariwal & Nichol (2021), our model is trained in 1.6M iterations and the baseline is trained in 4M iterations. Reference is the original data of ImageNet Deng et al. (2009)

Table 2: Comparing with state-of-the-art methods in ImageNet128 Deng et al. (2009), we use FID to evaluate the performance. Our method achieves the lowest FID score at each iteration.

| Methods | Iters=1M | Iters=2M | Iters=3M | Iters=4M |
|---|---|---|---|---|
| GD Dhariwal & Nichol (2021) | 17.18 | 8.14 | 6.63 | 6.04 |
| Min-SNR Hang et al. (2023) | 13.53 | 6.49 | 6.11 | 5.81 |
| GD+Ours | **7.24** | **5.91** | **5.48** | **5.40** |

Table 3: Comparing with state-of-the-art methods in Cifar10 Krizhevsky et al. (2009), we use FID to evaluate the performance. Our method achieves the lowest FID score at each iteration.

| Methods | Iters=100k | Iters=200k | Iters=300k | Iters=400k | Iters=500k |
|---|---|---|---|---|---|
| ID Nichol & Dhariwal (2021) | 5.40 | 3.48 | 3.05 | 2.72 | 2.60 |
| FDM Wu et al. (2023) | 4.91 | 3.03 | 2.58 | 2.49 | 2.43 |
| ID+Ours | **4.24** | **2.81** | **2.46** | **2.38** | **2.31** |

## 5.2 COMPARISONS WITH STATE-OF-THE-ART METHODS

We compare our optimized models with state-of-the-art methods, Min-SNR Hang et al. (2023) and FDM Wu et al. (2023). Table 2 compares the performance of our method with two state-of-the-art methods, Guided Diffusion (GD) Dhariwal & Nichol (2021) and Min-SNR Hang et al. (2023), on ImageNet128 Deng et al. (2009), and Table 3 compares with Improved Diffusion (ID) Dhariwal & Nichol (2021) and FDM Wu et al. (2023), on Cifar10 Deng et al. (2009). The results demonstrate that our method achieves the lowest FID score at each iteration of both datasets, indicating that our method outperforms the existing methods in terms of image generation quality and speed.

## 6 CONCLUSION

Our study focuses on the investigation of the stability and optimization of diffusion models. We provided empirical evidence to explain why diffusion models are stable and robust in learning the noise-to-data mapping, and in converging on the smoothing landscape. We also proposed enhancements to expedite the training of diffusion models by leveraging their distinctive properties: the curriculum learning-based timestep schedule and the momentum decay with learning rate compensation. The methods are evaluated on various models and datasets and demonstrate their effectiveness in enhancing the convergence speed and generation quality of diffusion models. Our work sheds light on the design and analysis of diffusion models and opens up new possibilities for improving their performance and efficiency.

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

## A PRELIMINARIES

We briefly review the definition of diffusion models from Ho et al. (2020). The diffusion process is a method of transforming the original data $x_0$ into random noise by adding Gaussian noise $\epsilon$ at each timestep $t$,

$$x_t = \sqrt{1 - \beta_t} x_{t-1} + \sqrt{\beta_t} \epsilon_t, \tag{11}$$

where $x_t$ is the noisy data at timestep $t$, $\beta_t$ is the noise level at timestep $t$. The diffusion process starts from $t = 0$ and ends at $t = T$, where $x_T$ is a pure noise vector. The noise level $\beta_t$ can be constant or vary across timesteps.

The reverse process is a way of recovering the original data from the noise by applying a denoising function $p(\cdot)$ at each timestep $t$,

$$x_{t-1} \sim p(x_{t-1}|x_t, t). \tag{12}$$

The reverse process starts from $t = T$ and ends at $t = 0$, $x_0$ is the generated data. The conditional distribution $p(x_{t-1}|x_t, t)$ is usually modeled by a neural network that outputs the mean and variance of a Gaussian distribution.

To train the neural network, we need to define a loss function that measures the discrepancy between the true distribution $q(x_{t-1}|x_t)$ and the learned distribution $p(x_{t-1}|x_t, t)$. One common choice is the KL divergence,

$$L = E_{x_0 \sim q(x_0), x_t \sim q(x_t|x_{t-1}), x_{t-1} \sim q(x_{t-1}|x_t)}[\log q(x_{t-1}|x_t) - \log p(x_{t-1}|x_t, t)], \tag{13}$$

where $q(x_0)$ is the data distribution, $q(x_t|x_{t-1})$ is the forward diffusion distribution, and $q(x_{t-1}|x_t)$ is the reverse diffusion distribution. The expectation is taken over all possible pairs of $x_t$ and $x_{t-1}$ for each timestep $t$. While Ho et al. (2020) further simplified the above loss function and the new loss function can produce better samples in practice,

$$L_{\text{simple}} = E_{x_0 \sim q(x_0), \epsilon \sim \mathcal{N}(0, \mathbf{I})} \left[ ||\epsilon - \epsilon_\theta(x_t, t)||^2 \right]. \tag{14}$$

To sample from the diffusion model, we need to sample a noise vector $x_T$ from a Gaussian distribution and then apply the reverse process to obtain $x_0$. This can be done by sampling $x_{t-1}$ from $p(x_{t-1}|x_t, t)$ at each timestep until $t = 0$. The final sample $x_0$ is the output of the diffusion model.

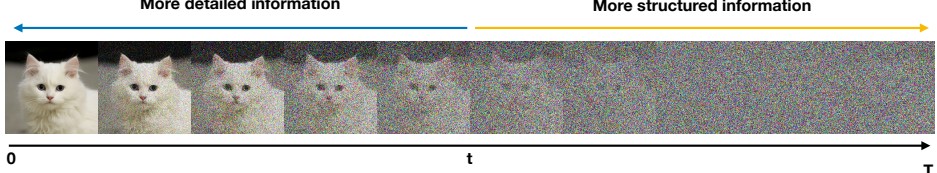

Figure 8: Visualization of the diffusion process.

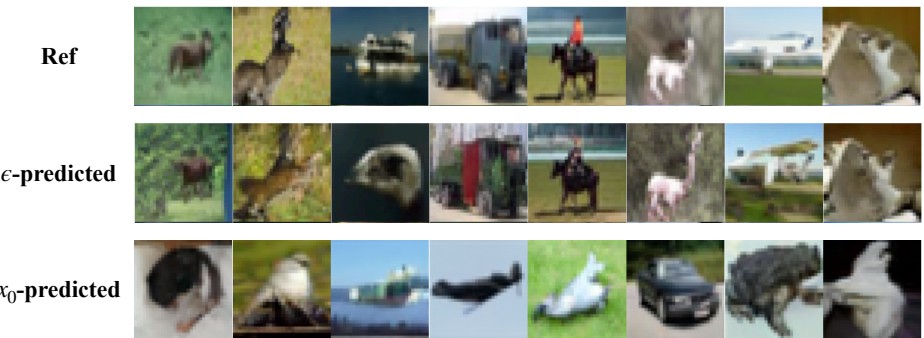

Figure 9: Illustration of consistency experiments of $\epsilon$-predicted and $x_0$-predicted DMs. The consistency phenomenon disappears when using $x_0$-predicted mechanism.

## B  EXPERIMENT SETTINGS

We trained our optimized models on Cifar10 Krizhevsky et al. (2009) and ImageNet128 Deng et al. (2009) datasets, following the hyper-parameter settings of Improved Diffusion Nichol & Dhariwal (2021) and Guided Diffusion Dhariwal & Nichol (2021), respectively. The details of the hyper-parameter settings are as follows:

For Cifar10, we used cosine timestep schedule, 4,000 timesteps, learning rate=1e-4, batch size=128. We used an exponential moving average (EMA) rate of 0.9999 for all experiments. We implemented our models in PyTorch Paszke et al. (2019), and trained them on $8\times$ NVIDIA 2080Ti GPUs, using 250 sampling processes. We used Adam optimizer, with $\beta_1 = 0.8, \beta_2 = 0.999$, which are based on the observation of the smooth landscape of diffusion models (DMs). The hyper-parameters of our proposed methods, namely curriculum learning based timestep schedule (CLTS) and momentum decay with learning rate compensation (MDLRC), are as follows: For CLTS, we set the mean value $\mu = 1200$ ($0.3 \times$ total timesteps, the optimized mean value through ablation study), and the target iteration=$5 \times 10^4$. For MDLRC, we set the lower limit of $\beta_1 > 0.4$.

For ImageNet128, we used linear timestep schedule, 1,000 timesteps, learning rate=1e-4, batch size=256. We also used an EMA rate of 0.9999 for all experiments. We implemented our models in PyTorch Paszke et al. (2019), and trained them on $8\times$ NVIDIA A100 GPUs, using 250 sampling processes. We used Adam optimizer, with initial $\beta_1 = 0.8, \beta_2 = 0.999$. The hyper-parameters of our proposed methods are as follows: For CLTS, we set the mean value $\mu = 300$ ($0.3 \times$ total timesteps), and the target iteration=$3 \times 10^5$. For MDLRC, we set the lower limit of $\beta_1 > 0.4$.

## C  RELATIONSHIP BETWEEN MOMENTUM DECAY WITH EXPONENTIAL MOVING AVERAGE

The iterative formula of AdamW is

$$m_t = \beta_1 m_{t-1} + (1 - \beta_1)g_t \tag{15}$$

$$v_t = \beta_2 v_{t-1} + (1 - \beta_2)g_t^2 \tag{16}$$

$$\hat{m}_t = m_t/(1 - \beta_1^t) \tag{17}$$

$$\hat{v}_t = v_t/(1 - \beta_2^t) \tag{18}$$

$$\theta_{t+1} = \theta_t - \gamma\hat{m}_t/\sqrt{\hat{v}_t} \tag{19}$$

The equation of EMA is

$$\hat{\theta}_{t+1} = \epsilon\hat{\theta}_t + (1 - \epsilon)\theta_t \tag{20}$$

As the decrease of $\beta_1$, the current gradient proportion increases, leading to the gradual dominance of the gradient in $m_t$, where $m_t$ is the main variable for updating $\theta_t$. Therefore, if the learning rate remains unchanged, the EMA model will be more influenced by the noise in the gradient.

## D  COMPARISONS OF LOSS LANDSCAPES

In this section, we comprehensively illustrate the loss landscapes of DMs and GANs across various training iterations, as shown in Fig. 10 and Fig. 11, respectively.

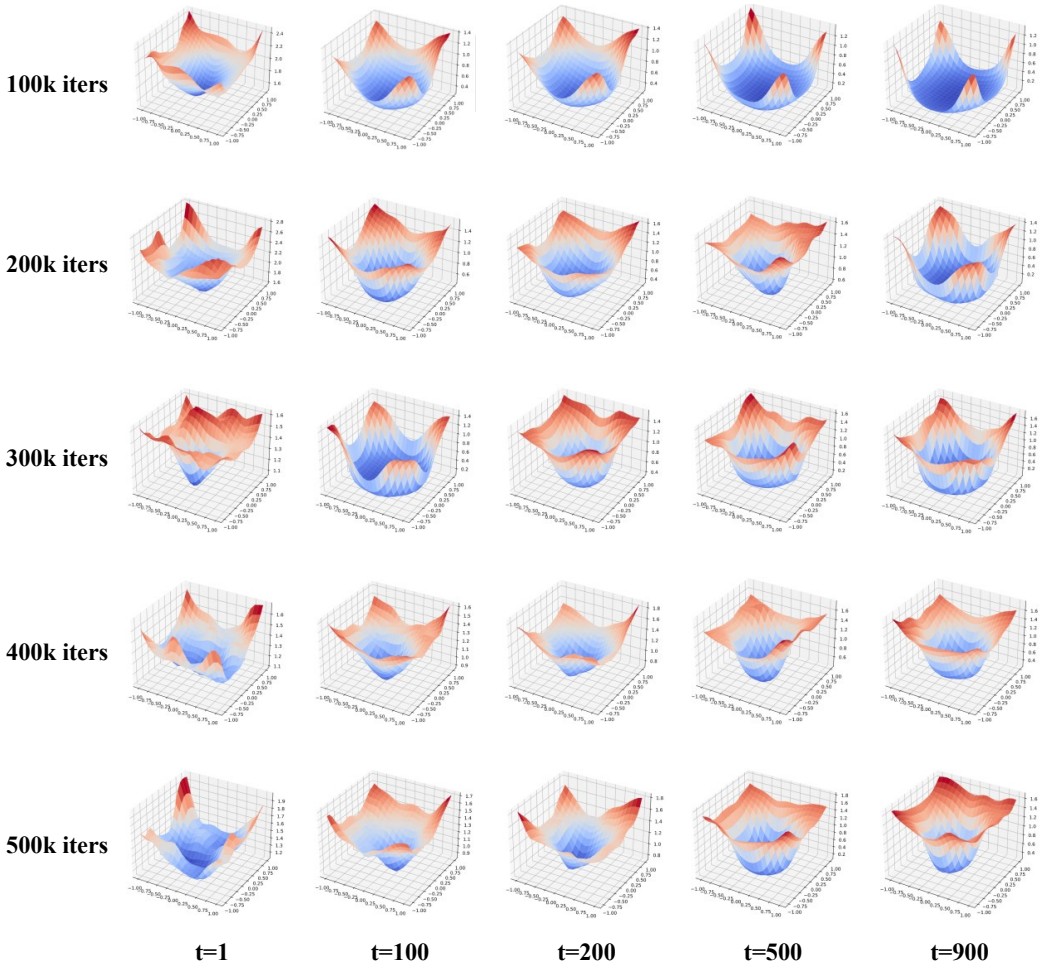

Figure 10: Loss landscape of diffusion models.

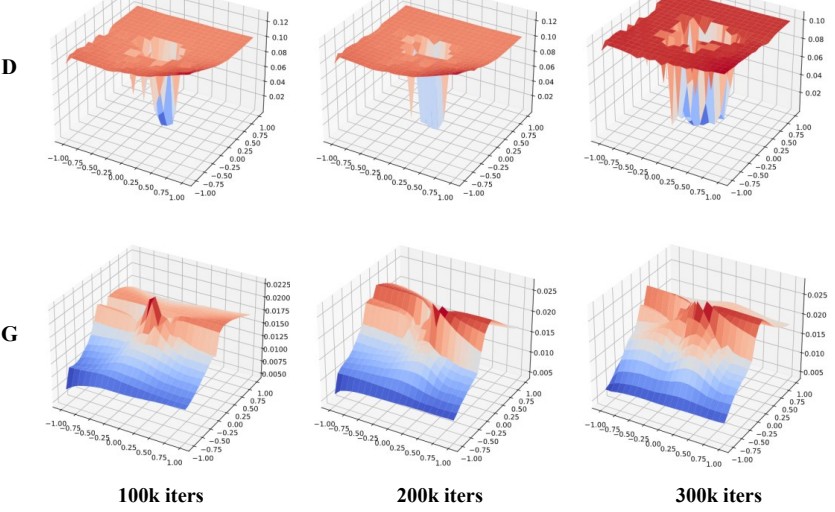

Figure 11: Loss landscape of generative adversarial networks.

# E  ILLUSTRATION OF CONSISTENCY EXPERIMENTS

In this section, we provide more results of consistency experiments among DMs and GANs of different frameworks. The results of models trained on cifar10, can be seen in Fig. 13, Fig. 12, and Fig. 14, from Improved Diffusion Nichol & Dhariwal (2021), Guided Diffusion Dhariwal & Nichol (2021), and DCGAN Radford et al. (2015), respectively.

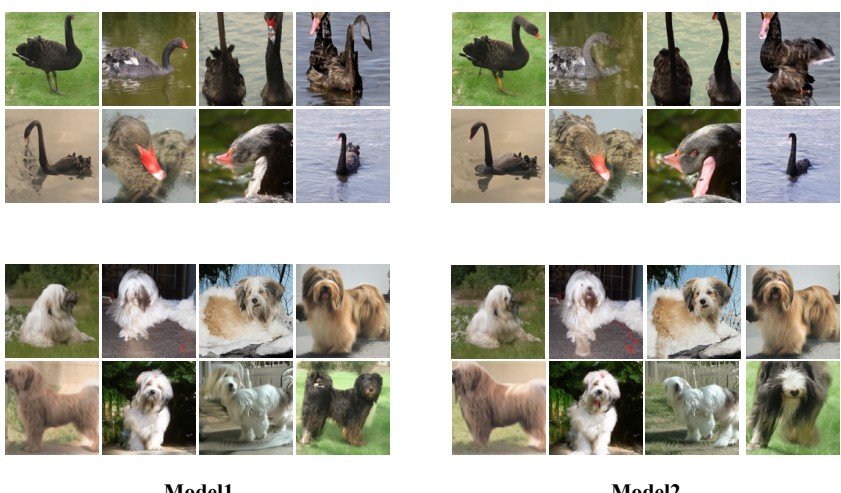

Model1                                        Model2

Figure 12: Consistency experiment results of Guided Diffusion at 128 resolution, each model trained on ImageNet. Obviously, DMs have a significant consistency phenomenon

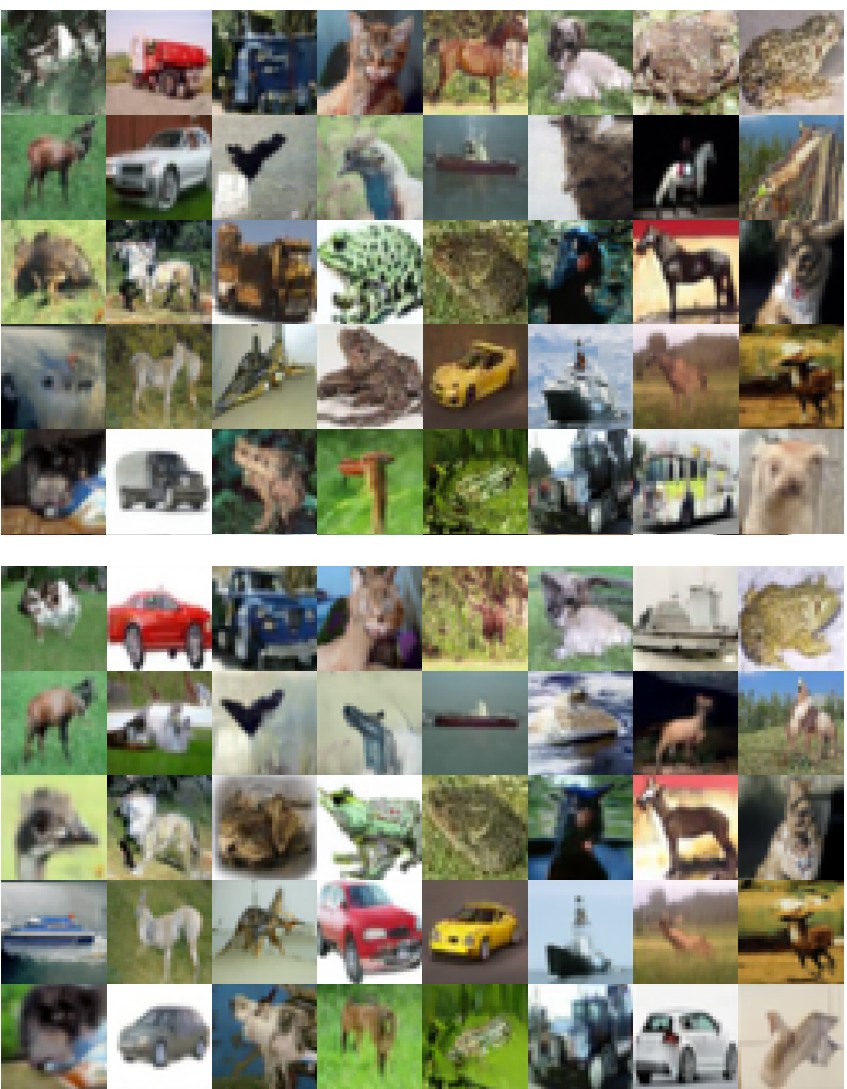

Figure 13: Consistency experiment results of Improved Diffusion at 32 resolution, each model trained on Cifar10.

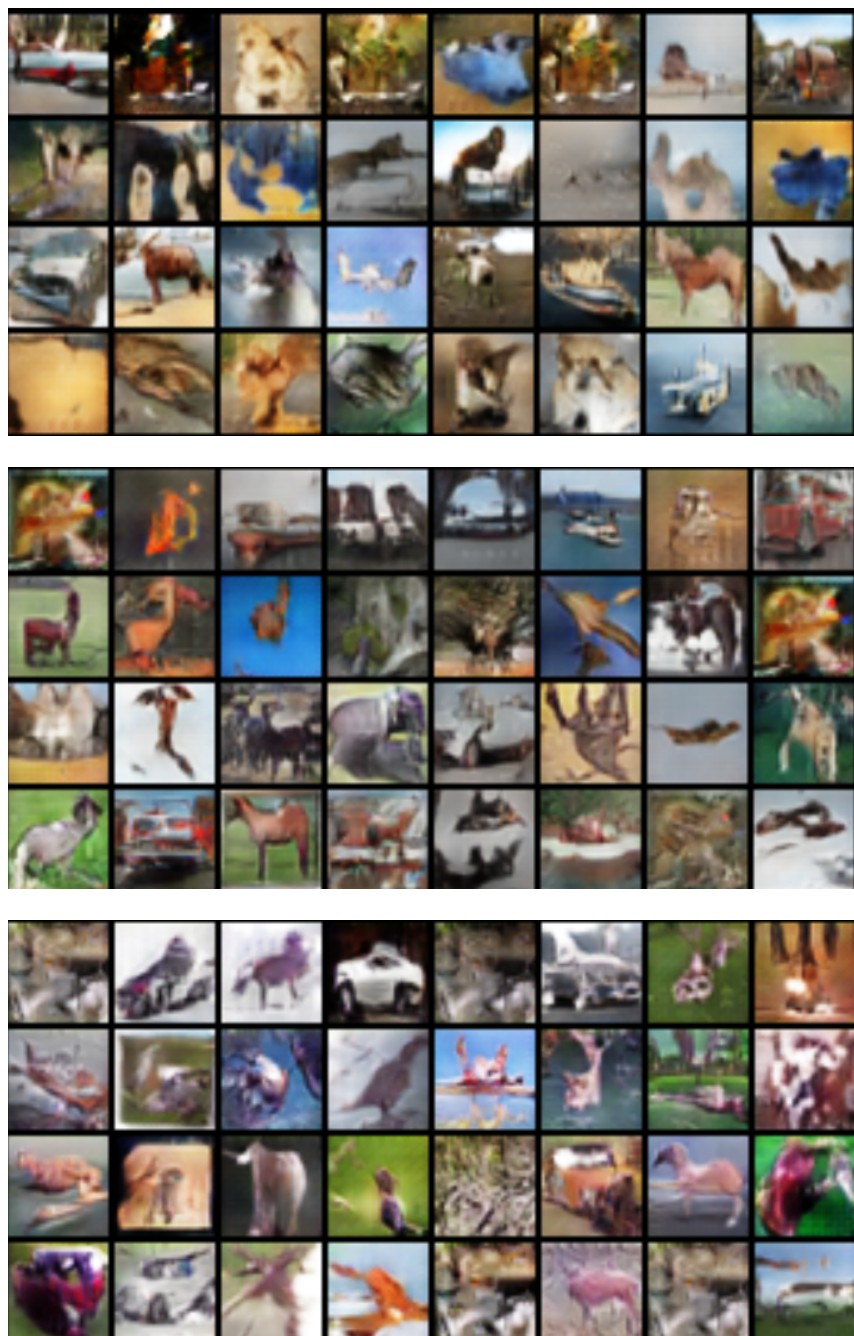

Figure 14: Consistency experiment results of DCGAN, three models with different initialization trained on Cifar10. Each model generates 32 images with the same sampling seed. Obviously, GANs have no consistency phenomenon.

# F ILLUSTRATION OF GENERATED RESULTS

we provide more generated results of our optimized DMs, as shown in Fig. 15.

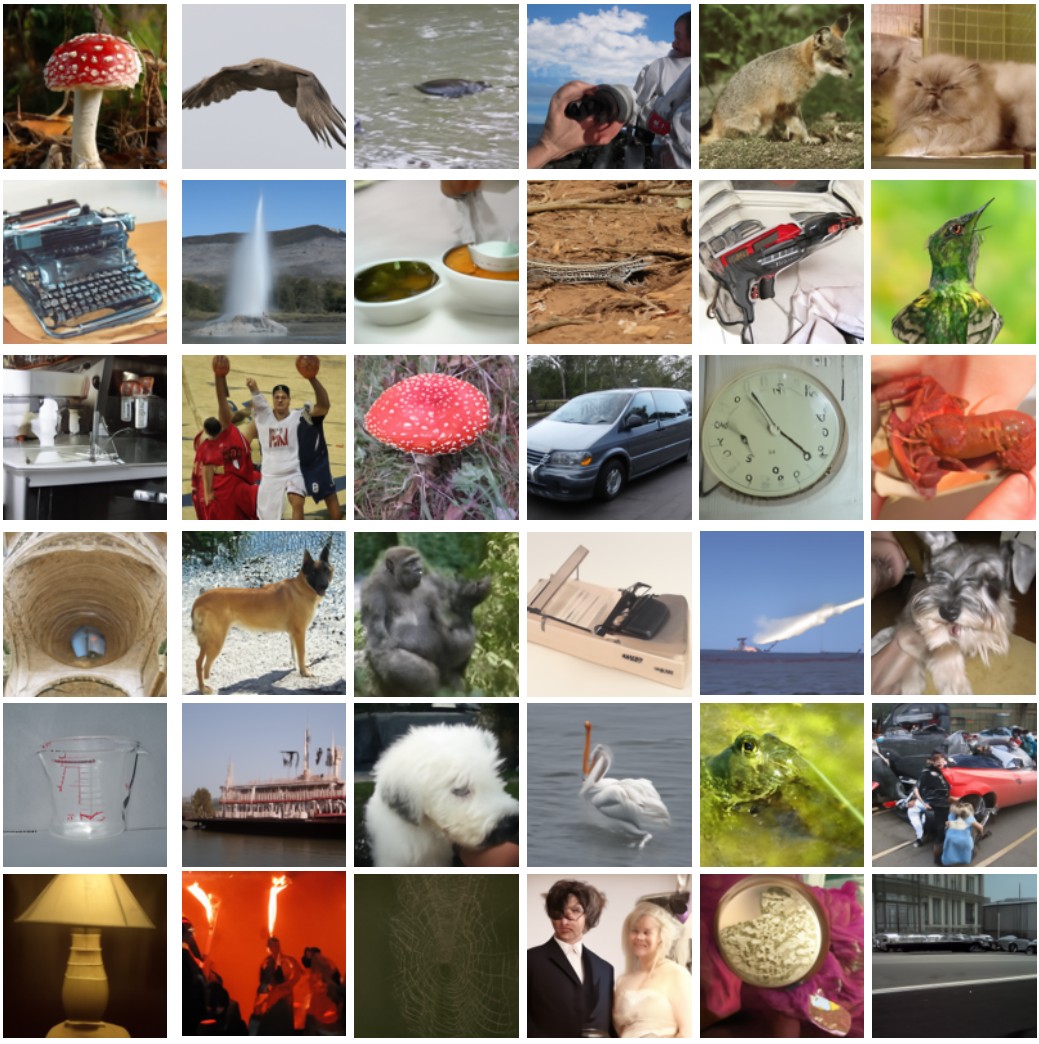

Figure 15: Results of our optimized diffusion model, which only trained 1.6M iterations on ImageNet128.

