# OpenReview forum: "Why Diffusion Models Are Stable and How to Make Them Faster: An Empirical Investigation and Optimization"
_ICLR.cc/2024/Conference — ICLR 2024 Conference Withdrawn Submission_

### Official Review · Reviewer_sw1d · 2023-10-31

**Soundness:** 2 fair
**Presentation:** 3 good
**Contribution:** 3 good
**Rating:** 5
**Confidence:** 4

**Summary:**

The paper studied the training stability of diffusion models compared to GANs. The paper showed that with the same sampling conditions, diffusion models produce more consistent generation results than GANs, across different training hyper-parameters such as initialization and model structure. The paper found that for \epsilon-prediction, timesteps close to T (which corresponds to generating high level structure) are easier to optimize than smaller timesteps, leading to consistent generation. Based on the observations, the paper proposed two optimization techniques for diffusion models, curriculum learning based timestep schedule (CLTS) and momentum decay with learning rate compensation (MDLRC), that are demonstrated to enable more efficient training with up to 2.6 speedup.

**Strengths:**

1. The paper provided experimental evidence on why the training of diffusion models is more stable than GANs. These results are valuable for understanding and improving diffusion models.

2. The paper proposed two effective techniques to improve the training efficiency of diffusion models, and well explained their motivation and formulation. The techniques look promising in experiments on different datasets, ImageNet128 and Cifar10.

**Weaknesses:**

1. The paper did not provide sufficient details of the consistency experiment, which is crucial to the paper. Figure 1 shows that different diffusion models generate surprisingly similar images, even when model structures are different. More experimental details can better support these findings. See Question 1.

2. The paper could provide more concrete implications of the stability phenomenon. Firstly, it seems that there is some connection between stability and overfitting (or memorization). Table 1 showed that the consistency score is higher on smaller images, and Figure 9 showed that the \epsilon-predicted model can generate images quite similar to Ref. Does this mean diffusion models are easier to overfit? Secondly, the consistency experiment showed that the training is insensitive to certain hyper-parameters and model structures. More concrete guidelines from these results could be useful such as what configs are important and should be carefully tuned for training diffusion models, and what are not.

**Questions:**

1. In consistency experiment (Section 3.1), what are the different initializations? Does the initialization use different random seeds or different distributions? What are the different model structures considered? And how large is N?

2. For ablation study in Section 5.1, how dow CLTS alone perform? Moreover, if CLTS improves training efficiency, can it work as a better scheduler for inference too, e.g., by focusing on more important timesteps during sampling?

3. What is the impact of CLTS and MDLRC on the stability phenomenon? Results like consistency scores and visual similarity comparison for these models would be interesting.

4. How is stability related to overfitting? [1] discussed the overfitting-generalization trade-off in VQGAN. In Appendix.E they showed that on face datasets, early-stopping can prevent overfitting and generate new faces, while longer training can reproduce faces in the training set and get smaller FID. Does this conclusion hold for diffusion models on small datasets as well?

5. Does the stability phenomenon exist in text-to-image diffusion models (such as Stable Diffusion) as well?

[1] Esser, Patrick, Robin Rombach, and Bjorn Ommer. Taming transformers for high-resolution image synthesis. CVPR 2021. https://arxiv.org/pdf/2012.09841.pdf

---

### Official Review · Reviewer_fHcz · 2023-10-31

**Soundness:** 3 good
**Presentation:** 3 good
**Contribution:** 2 fair
**Rating:** 3
**Confidence:** 4

**Summary:**

This study elucidates that the training stability of diffusion models stems from the noise-to-data mapping's stability and the smoothness of the loss landscape. Building on these insights, the research introduces a Curriculum Learning based Timestep Schedule (CLTS) and Momentum Decay with Learning Rate Compensation (MDLRC), effectively doubling the training speed of diffusion models.

**Strengths:**

(1) The paper is clearly-written with experimental results supporting the claims.

(2) The acceleration techniques introduced in the study successfully enhance the training speed of diffusion models, surpassing prior methods.

**Weaknesses:**

(1) The paper's claims, while supported by experimental results, lack robust theoretical analysis. This reliance on experimental evidence alone may render the claims less persuasive, as experimental outcomes can sometimes be situational or coincidental.

(2) The proposed CLTS acceleration technique appears conceptually similar to progressive noise schedules, which incrementally adopt denser noise schedules over crucial timesteps for generation—a concept employed in other works [1]. Additionally, the momentum decay technique does not emerge as a novel approach in the training of diffusion models.

[1] Song, Y., Dhariwal, P., Chen, M. and Sutskever, I., 2023. Consistency models.

**Questions:**

(1) In equation (5), why $\epsilon_{\theta} \rightarrow I$, as $\epsilon$ is a Gaussian random variable with mean $0$?  $\epsilon_{\theta}$ should approach zero.

(2) Can the author elaborate more about how to tell from Figure 8 that the diffusion model generates structural information when $t\rightarrow T$?

---

### Official Review · Reviewer_bVYL · 2023-11-01

**Soundness:** 1 poor
**Presentation:** 3 good
**Contribution:** 1 poor
**Rating:** 3
**Confidence:** 4

**Summary:**

The authors demonstrate that diffusion models (DMs) consistently produce similar generation results across varying hyper-parameters. Specifically, they find that diffusion models with different architecture and initialization parameters generate similar images when the initial noise and noise injected at each step remain the same. The authors attribute this consistency to the noise prediction from noisy images in DM learning.
Additionally, the paper explores the stability of DMs through the application of 1D interpolation and Hessian spectra techniques. The analysis reveals that the loss landscapes of DMs are notably smoother than those of GANs.
Based on these findings, the authors propose two optimization methods for diffusion models: the curriculum learning-based timestep schedule (CLTS) and the momentum decay with learning rate compensation (MDLRC). They argue for CLTS is justified by the observation that epsilon prediction approaches an identity function as timestep (t) approaches the total number of steps (T), making it easier to train for t->T, allowing for a curriculum that starts from easier task (t near T) to more challenging ones (t near 0). MDLRC is proposed to address potential oscillation during training caused by large momentum values and favorable loss landscapes of DMs.

**Strengths:**

The paper is well-structured and easy to follow.

**Weaknesses:**

Weaknesses are written in following parts.

**Questions:**

The stability analysis appears an application of existing methods to DMs, lacking novel contributions in this aspect.

The claim of consistency in DMs may be considered trivial, as for the same training data, the target function with minimal loss is unique, or denoising models (not diffusion) produce similar clean images while varying hyperparameters.

The statement that mean-prediction model does not have consistency require further clarification and evidence. For example, if we choose the architecture with skip-connection of $1/\alpha_t x – \sqrt{1/\alpha_t-1} f_\theta(x,t)$, mean prediction is equivalent to epsilon prediction with the architecture $f_\theta(x,t), thus the consistency might be a matter of architectures.

The paper lacks a comprehensive evaluation of the proposed MDLRC, with only two experiments presented, compromising the credibility of experiments.

Several conclusions in the paper remain unexplained or unexplored..
- Section 4.2 The consistency phenomenon indicates … are easy to converges.: lacks a clear rationale, as for the epsilon prediction is almost identity does not imply easy convergence. Reverse conclusion is indeed more plausible for me: since denoising is difficult for large noise, timesteps near 0 are easy to converge.
- Section 4.3 Applying a large momentum value in DMs … but also cause oscillations,: How large momentum optimization algorithm on a smooth landscape affect convergence efficiency? What oscillations mean?
- Section 4.3 simply applying momentum decay … over amplification … stability of the EMA model.: Why momentum decay cause over amplification and how over amplification affect stability?
- Section 5.1 The results suggest that the optimal value of … the most difficult ones to learn.: require additional arguments to support significance of timesteps around 0.3, as the stated results only indicate the superiority of the proposed timestep scheduling with mu 0.3 over ones with other values.